# Dynamic Feedback versus Varna-Based Techniques for SDN Controller Placement Problems

**Wael Hosny Fouad Aly** [1,*], **Hassan Kanj** [1], **Samer Alabed** [1], **Nour Mostafa** [1] **and Khaled Safi** [2]

1 College of Engineering and Technology, American University of the Middle East, Egaila 54200, Kuwait; hassan.kanj@aum.edu.kw (H.K.); samer.al-abed@aum.edu.kw (S.A.); nour.moustafa@aum.edu.kw (N.M.)
2 Computer Science Department, Strasbourg University, 67081 Strasbourg, France; ksafi@unistra.fr
* Correspondence: wael.aly@aum.edu.kw

**Abstract:** During the past few years, software-defined networking (SDN) has become a successful architecture that decouples the control plane from the data plane. SDN has the capability to monitor and control the network in a central fashion through a softwarization process. The central element is the controller. For the current SDN architectures, there is an essential need for multiple controllers. The process of placing the controllers efficiently in an SDN environment is called the *controller placement problem (CPP)*. Earlier CPP solutions focused on improving the propagation delays through the capacity of the controllers and the dynamic load on the switches. In this paper, we develop a novel algorithm called *dynamic feedback algorithm for controller placement for SDN* ($DFBCP_{SDN}$). $DFBCP_{SDN}$ is compared with the *varna-based optimization (VBO)* towards solving the CPP. We used the VBO as the reference model to this work since it is relatively a new algorithm. Moreover, the VBO extensively outperformed many other existing models. To the best of our knowledge, this is one of the first attempts to minimize the total average latency of SDN using feedback control theoretic techniques. Experimental results indicate that the $DFBCP_{SDN}$ outperforms the VBO algorithm implemented in two well-known topologies, namely *Internet2 OS3E topology* and *EU-GÉANT topology*. We observe that for uncapacitated CPP, the $DFBCP_{SDN}$ outperforms the VBO for Internet2 OS3E and EU-GÉANT topologies by 11% and 9%, respectively, in terms of total average latency. On the other hand, for capacitated CPP, the $DFBCP_{SDN}$ algorithm outperforms the VBO reference model by 10% and 8%, respectively.

**Keywords:** SDN; feedback control theoretic; controller placement; latency; varna-based optimization; ARMA models

## 1. Introduction

Software-defined networking is an emerging paradigm which provides a separation between the control plane and the data plane [1–7]. The data plane is responsible for forwarding the traffic based on the controllers' decisions. The network traffic is handled by the control plane. The control plane is responsible for producing sufficient rules and policies to the forwarding devices, whether these forwarding devices are switches or routers [8–11]. When a new flow is sent to the switch, the switch sends a specific message to its corresponding controller to setup the flow rules along with the best flow path. The controller in charge manages the routing of flows through interacting with switches securely. A control plane guides the switches on how packets should be forwarded by configuring new flow rules and policies [12].

In a WAN environment, a single controller is not sufficient to handle the entire overload of switches that are physically distributed, since it cannot guarantee the desired latencies among switches and controllers. A server has limited capacity in order to handle a large number of messages generated by its associated switches [13–16]. As a result, SDN-based WANs use multiple controllers to increase the network's performance. Hence, in this work, we assume multiple controller hierarchy. The controller placement problem (CPP)

is not a trivial problem. Placing a controller should be based on crucial metrics such as average delay between the switches and the controller, maximum delay between the switch and the controller, and intercontroller latency. The capacity of the controller reflects the number of incoming packets processed by the controller per second. CPP is categorized into two subcategories: (1) the capacitated controller placement problem (CCPP) and (2) the uncapacitated controller placement problem (UCPP). The uncapacitated category refers to infinite capacity buffered controllers; on the other hand, the capacitated category refers to controllers having finite capacities [17–19]. The controller's capacity refers to the processing rate of the incoming messages. Controllers in the uncapacitated category are assumed to have unlimited packet processing power and are capacity-independent. On the other hand, the controllers in the capacitated category could have either equal or unequal capacities [20].

The performance of the controller [21] is a crucial factor to achieve the desired scalability. Heller et al. [22,23] discussed the CPP as a facility location problem which is considered as an NP-hard problem. To solve the CPP, researchers studied the CPP as a partitioning problem where large networks are partitioned into smaller network clusters where one controller is hired per cluster [24]. Network partitioning reduces the overall complexity for complicated networks. Gao et al. [25] introduced a node swarm optimization (PSO) algorithm to minimize the total average latency of the network. Propagation delay is the time taken for packets to travel between any two network elements (switches or controllers). Usually, propagation delays are measured in terms of hops or distances between the network nodes, whether the network element is a controller or a switch.

If a switch, associated with a controller, sends a message to another switch associated with another controller, this is achieved through intercontroller communication (ICC) [26,27]. ICC is implemented using *border gateway protocol* (*BGP*) [28]. The larger the number of the controllers, the more complicated the intercontroller communication needed to handle the network to achieve an improved overall network performance. Therefore, minimizing the total number of controllers plays an important role towards increasing the SDN's performance. Although assigning one controller per switch minimizes the switch-to-controller propagation delay, it also leads to an increase in the intercontroller communication overhead. This leads to a reduction in the controller's utilization [21]. Most of the research on controller capacity limitation focuses on both the performance and the capacity of the SDN controllers [29,29–31].

An example of the capacity limitation is the *c-bench* [32,33]. The c-bench is a simulator tool that is used to measure the number of controller flows per second. The SDN controller is able to control a set of switches due to the limited available resources. The NOX controller is an example of a controller that can handle up to 30,000 flows setups per second. In such cases, load balancing among controllers is required to increase the performance of SDN. The switch workload could handle a certain number of incoming and outgoing requests. An overloaded switch could drop packets and, hence, degrade the overall SDN performance. The overall nodal delay is the total switch-to-controller delays in addition to the intercontroller delays. In general, researchers seek to maximize the SDN's performance by minimizing the total nodal delay of the network [11,12,25,34–39]. To the best of our knowledge, there is no approach to consider all the above factors at once for a given solution to the controller placement problem. In this paper, we tackle different scenarios for CCPP and UCPP including different distances among various network elements and also the workloads measured at switches. We consider both cases, whether the weights are the same and different. The weights on the links between any two network components reflect the propagation delay between these components.

This paper is organized as follows: Section 2 has the related work. Section 3 has the mathematical model for controller placement problem. Section 4 has the VBO reference model. Section 5 has the proposed model DFBCP$_{\text{SDN}}$. Section 6 has the results and analysis. Section 7 has the conclusion and the future work.

## 2. Related Work

This section provides a short overview of selected literature that deals with the controller placement problem. We noticed that work in [18,40–43] does not take the capacities of the controllers and switches into consideration. Static environments assume that the burden on the controllers could not be distributed to other underutilized controllers since those environments have fixed traffic. Traffic is assumed to be measured in terms of propagation delay, the hop number, or a number of packets transferred between switches and controllers. Heller et al. [18] has modeled the CPP through clustering the controllers into groups of controllers in order to minimize the average latency at switches and controllers during the controller placement process. The authors in [18] modeled the CPP problem using the minimum k-median model. The k-median problem is considered an extension to the k-mean problem where it divides the network into k-clusters; each cluster computes the median rather than the mean. The work in [18] focused on only static environments that ignore the workload at the controllers, controller–controller latency, and various failure cases. It is intuitive that a controller might fail if the workload is not taken into consideration. Load balancing is crucial in such scenarios. The capacity of the controller refers to the rate of the arriving packets.

Another direction towards reducing the CPP effect is through optimization. Optimization techniques have been used by several research groups such as Sallahi et al. [20,44]. The model used by Sallahi el al. [44] depends on activating and deactivating controllers and links to improve the overall performance of the network using the CPLEX optimizer to find the optimal number of controllers to minimize the cost through a mathematical model. The problem with the proposed model is that it is only applicable for small areas. In the literature, the clustering-based techniques were addressed on many occasions [45–50]. Moreover, optimization-based techniques were also addressed in research by numerous authors [51–54]. The work in [51,52,55–57] has studied the workload factor in addition to the weights among various switches of the network. The partitioning of large scale network into smaller domains first started via the FlowVisor architecture [58]. FlowVisor uses multiple controllers, where each controller manages a domain. Liao et al. [55] proposed clustering the large-scale SDN network into smaller domains, where one controller manages each partitioned domain. Density-based controller placement has outperformed k-center for the CPP [59]. Previous work in [25,51,52,56,60] discussed the CPP solutions through heuristic-based optimization techniques. Gao et al. [25] proposed a node swarm optimization (PSO) algorithm to solve the CPP based on heuristic solutions. In the proposed solution, the authors were able to find the optical placement, but the algorithm proposed needed to know the number of controllers in advance. Sherwood et al. [58] proposed the FlowVisor model that was one of the first models to introduce utilizing multiple controllers for SDN architectures. The proposed model is based on splitting the large-scale network into several clusters, where each cluster is controlled and administered by a single controller.

Density-based clustering is one of the contributions that is based on the partitioning techniques proposed by Liao et al. [55]. Liao et al. [55] have used clustering that is based on density to divide the large-scale SDN network into smaller clusters that are administered and controlled by a controller. The authors in [55] proved that density-based controller placement outperformed the k-center technique proposed by Yao et al. [59]. According to the literature, the CPP is considered to be an NP-hard problem. The work in [25,51,52,56,60] discussed the CPP and provided heuristic and optimization solutions for it. Gao et al. [25] introduced a node swarm optimization-based algorithm to solve the CPP. The authors in [25] found optimal controller placements, but the proposed techniques assume to know the number of controllers in advance. The controller's capacity parameter was not studied with the effect of the loads on the switches. The work proposed by [51,52,55–57] has considered both the controllers' capacities and the switches' workloads. The proposed work has only taken the unit weight among the switches of the network into consideration.

The work proposed by Mohanty et al. [61] developed a model that depends on optimization techniques to reduce the overall latency. The authors in [61] also claimed to find the adequate number of controllers while maintaining the minimized overall latency. The main disadvantage of the proposed approach is that load balancing among controllers is not guaranteed and, hence, overloaded controllers might fall, and hence the model needs to recover from cascaded failures through fault tolerance techniques.

The work proposed by Huang et al. [62] developed a genetic-based algorithm and gradient descent optimization technique. The authors used the genetic-based algorithm to search for suitable CPP solutions, while the gradient descent was used for evaluation purposes. The drawback of the proposed model is the high complexity of the model due to the time utilized by the genetic algorithm to search for the suitable controller, and hence the algorithm is not considered as a transparent algorithm due to its overhead. The authors also focused on the control plan utilization in the proposed model and did not study the sufficient workload on the controllers that might lead to overloading the controllers, and hence might require the fault tolerance algorithm to recover from cascading failures scenarios as well.

Tao et al. [63] focused on the total flow request cost through considering the switches' weights, switching to controller routing costs, and intercontroller routing costs. The main focus of the work proposed by Tao et al. [63] is to balance the load among controllers for a fixed number of controllers. In order to find the position of the controllers, the authors used the minimization of the linear function of the load balance factor in addition to the total flow request. In this work, latencies are not taken into consideration for decision making, and hence this approach is considered slow and might lead to low-performance decisions for the CPP solutions.

In this paper, we study both the unit-weighted edges among switches and the controller's capacities. We suggest to use the propagation delays as weights based on modifying the density-based controller placement algorithm. The switch workload is measured as the packets sent to the controller per second. We use different scenarios for the workloads at the switches. We categorized the workloads into same/different workloads. Same/different workloads indicate that the switches generate messages with the same/different rates to their associated controllers. If the associated controllers have the same or different capacities, this is considered an indication that the controller has equal or different packet processing power, respectively.

## 3. Mathematical Model for the Controller Placement Problem

In this work, we assume that the network topology is represented as a graph of the tuple $G(S; E)$, where $S$ represents the set of switches and $E$ represents the set of edges connecting switches to their corresponding controllers. We started by analyzing the network topology and then cluster the network into smaller partitions. A controller is deployed per cluster. Due to the increase in the demand of the daily traffic, one controller is not sufficient to handle such traffic. The capacity of the controller and the workloads on the switches are crucial elements for decision making. When we consider the capacity of the controllers and the workloads on the switches, our proposed model outperformed other techniques. The proposed algorithm is called dynamic feedback control theoretic algorithm ($DFBCP_{SDN}$). $DFBCP_{SDN}$ is compared with a well-known reference model that uses the varna-based optimization (VBO) to solve the CPP. The average switch-to-controller delay is computed by Equation (1),

$$\prod^{avgSW2CTRLdelay}(CP) = \frac{1}{n} \sum_{sw \in S} \left( \min_{cp \in CP} \right) d(sw, ctrl) \tag{1}$$

The maximum switch-to-controller delay is represented by Equation (2).

$$\prod^{maxSW2CTRLdelay}(CP) = \min_{CP \subseteq SW} \max_{sw \subseteq SW} \min_{CP \subseteq CP} \tag{2}$$

The average controller-to-controller delay is provided in Equation (3),

$$\prod^{avgCTRL2CTRLdelay}(CP) = \frac{1}{p_{inter}} \sum_{ctrl_i, ctrl_j \in CP} d(ctrl_i, ctrl_j) \tag{3}$$

The maximum controller-to-controller delay is provided in Equation (4),

$$\prod^{maxCTRL2CTRLdelay}(CP) = \max_{ctrl_i, ctrl_j \in CP} d(ctrl_i, ctrl_j) \tag{4}$$

where:

- $d(sw, ctrl)$ is the shortest path between switch $sw \in SW$ and controller $ctrl \in CTRL$.
- $d(ctrl_i, ctrl_j)$ is the shortest path between controllers $ctrl_i, ctrl_j \in CP$.
- $CP \subseteq SW$ is the set of all possible placements for controllers.
- $p_{inter}$ is the total number of intercontroller paths.

The total average delay is computed by Equation (5).

$$Total^{avg-delay}(CP) = \frac{1}{n} \sum_{sw \in SW} \min_{ctrl \in CP} d(sw, ctrl) + \frac{1}{p_{inter}} \sum_{ctrl_i, ctrl_j \in CP} d(ctrl_i, ctrl_j) \tag{5}$$

The total maximum delay could be computed by Equation (6).

$$Total^{avg-delay}(CP) = \min_{CP \subseteq SW} \max_{sw \in SW} \min_{ctrl \in CP} d(sw, ctrl) + \max_{ctrl_i, ctrl_j \in CP} d(ctrl_i, ctrl_j) \tag{6}$$

The main purpose of this work is to minimize the total average delays which could be mathematically modeled by Equation (7). Equation (8) guarantees that the overall loads on the switches do not exceed the capacity of the associated controller.

$$minTotal^{avg-delay}(CP) \tag{7}$$

Subject to:

$$\sum_{sw \in SW(ctrl)} wl(sw) \leq WL(ctrl), \forall ctrl \in CP. \tag{8}$$

where $WL(ctrl)$ is the workload capacity of the controller and $wl(sw)$ is the workload at the switch $sw$.

## 4. Reference Model: Varna-Based Optimization (VBO) for CPP

This section discusses the varna-based optimization (VBO) reference model [20], which we used in this research article as the reference model. The reason we used the VBO as the reference model is that the VBO is a relatively new approach that addresses the CPP. In addition to this, it was compared with numerous numbers of other models. The term varna indicates a class. The VBO tackles the CPP through deciding how many controllers are required for optimal placement. The VBO considered different scenarios for different workloads at the switches and capacities at the controllers.

The weight of the links between two switches reflects the delay between them. Different weights are studied in the experiments conducted in the paper. Using the same weight links indicates that the links have the same number of hops, while different weights indicate that the links have various delays. The load on the switch is computed as the number of packets sent to their corresponding controller per second. Controller capacity reflects the number of messages processed by the controller per second.

In the VBO algorithm, nodes are classified into two *classes*. The two classes are assumed to be (*Class A* and *Class B*). The classes are chosen based on the capability of the nodes. Nodes that have higher capabilities belong to *class A*, while nodes that have lower capabilities belong to class B. The remaining nodes are classified as *class B* nodes. *Class A* nodes are able to use the best route and refrain from using the worst routes. *Class B* nodes are able to interact with other nodes through peer communication. Class sizes are assumed to be $\alpha$ for *class A* and the remaining nodes are for *class B*, where $\alpha$ is the fraction of the entire population. The authors in [20] assumed that $\alpha$ could take values from 0.05 to 0.2. The value of $\alpha$ used is 0.10 for experimentation purposes. The authors used the peer constants $c_1 = 1.50$ and $c_2 = 1.25$. These values are used to control the search regions among alternatives. The value of $c_1$ is chosen to be higher than $c_2$, since this value gives a higher chance of a promising solution around a node that has the best solution. The authors in [20] assumed that *class A* nodes move towards the best route while moving away from the worst route, as given by Equation (9).

$$X_{i'} = X_i + r_A \times (X_{best} - X_{worst})$$ (9)

For every node $X_i$ in class B, the VBO randomly selects a node from the whole population, referred to as $X_{peer}$.

If the capability of the node $X_i$ is better than $X_{peer}$, the VBO moves that node towards the best solution and away from the peer solution, as shown in Equation (10)

$$X_{i'} = X_i + c_1 \times r_B(X_{best} - X_{peer})$$ (10)

If the capability value of the node $X_i$ is worse than that of $X_{peer}$, the VBO moves the node towards $X_{peer}$, as shown in Equation (11)

$$X_{i'} = X_i + c_2 \times r_B(X_{peer} - X_i)$$ (11)

If both nodes have the same capability value, then the new position is updated as twice that of the current position, as shown in Equation (12)

$$X_{i'} = 2 \times r_B \times X_i$$ (12)

The VBO performs the following: (1) Calculates the positions of the new nodes for class A using nodes with minimum and maximum latency. (2) Calculates the positions of the new nodes for class B by using latencies of peer nodes. (3) Calculates the latency of each new node using objective function $f$. (4) Compares the latency of the new node with the latency of the old node. (5) Finds the optimum placements.

## 5. Proposed Model: DFBCP$_{SDN}$ for CPP

This section has the proposed model referred to as dynamic feedback model for controller placement (DFBCP$_{SDN}$). DFBCP$_{SDN}$ uses feedback control theoretic techniques for placing the controller for SDN networks. DFBCP$_{SDN}$ calculates the utility function to appropriately place the controller. DFBCP$_{SDN}$ takes the placement decision based on a set of parameters such as the *propagation delay* and *percentage of packet losses*. DFBCP$_{SDN}$ controller placement depends on the average workload at the controller and the number of switches associated with each controller.

The block diagram of the feedback control system used in this paper is shown in Figure 1. The proposed model is referred to as DFBCP$_{SDN}$. DFBCP$_{SDN}$ determines the controller placement selection through feedback ARMA approaches. The actuator module detects the proper place for each controller. If the utility function returns a value that is different than the target reference value, the range of the accepted values, the controller's control law takes a proportional action to determine the controller placement. As shown in Figure 1, a system under control is distinguished by a parameter called the *controlled output*

*parameter*. The control output parameter (COP) is a parameter we seek to achieve a certain value for, but the problem is that the COP cannot not be accessed in a direct way.

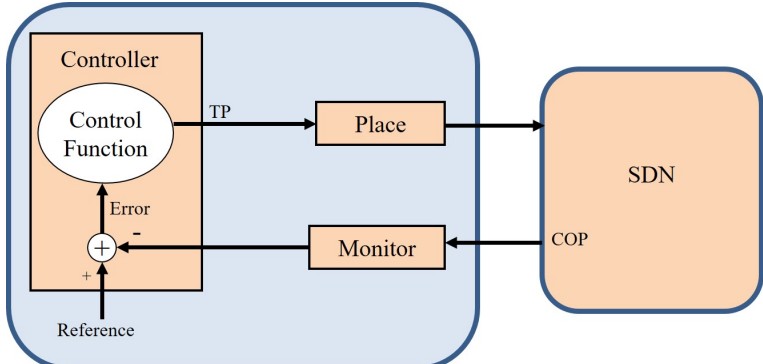

**Figure 1.** Block diagram for Feedback Control System for the DFBCP$_{SDN}$ model.

On the other hand, a tuning parameter (TP) is the parameter we use to tune the controlled output parameter in order to achieve the target reference value. TPs have direct impacts on the COP. The TP is a parameter that could be manipulated and updated directly by an administrator. The TP has an influence and impact on the COP through the feedback mechanism. DFBCP$_{SDN}$ uses the controller placement index (CPI) as the COP. The CPI identifies the location of the controller. The COP sends a feedback of the sensor module that reads the COP and compares it with the target value. The difference in the values between the reference value and the COP read value constitutes the error value. The error value is used as an input to the control law module placement engine module. Based on the error value, the tuning parameter value is computed. The error value affects the controller's output decision through the control law engine.

Figure 2 has the feedback control system for the DFBCP$_{SDN}$ model. The *controller engine* uses the control function that updates the tuning parameter. The *place module* is responsible for executing the appropriate placing algorithm based on ARMA approaches. The *monitor module* is used to detect the current value of the utility function of the SDN. The output is fed back to the comparator to be compared with a specific reference value. The value of the reference value is assumed to be 90% in this model. DFBCP$_{SDN}$ functions in two phases, as shown in Figure 3: (1) The *system identification phase*, which is responsible for modeling the system mathematically, and (2) the *control law phase*, which is responsible for detecting the stable regions for selecting the roots of the controller.

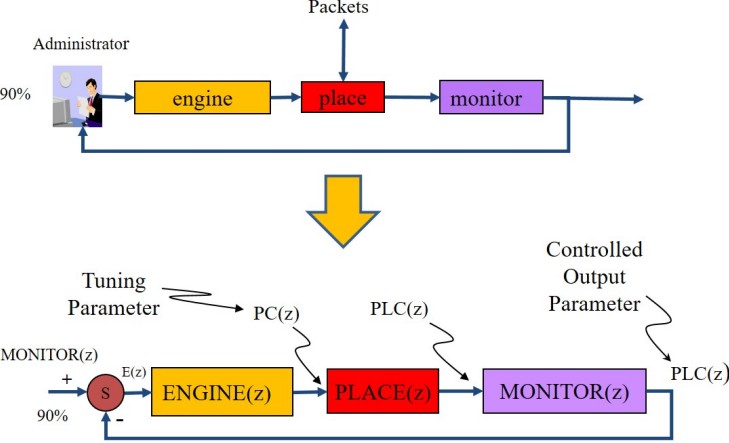

**Figure 2.** Feedback control for SDN networks.

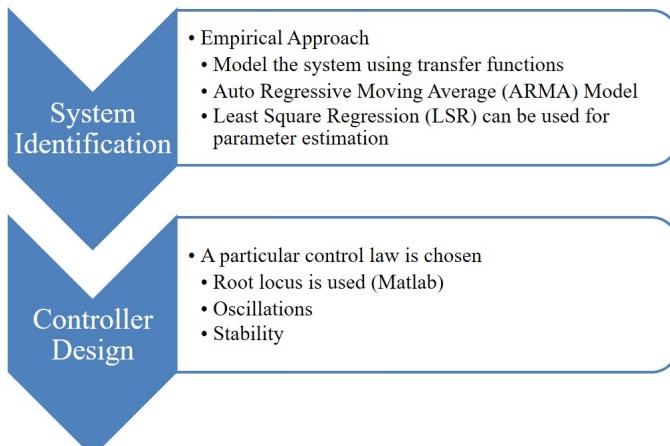

**Figure 3.** Classical control theory design.

DFBCP$_{PC}$ uses the weighted utility function shown in Equation (13)

$$\text{DFBCP}_{PC} = \varpi_1 * \tau + \varpi_2 * \rho + \varpi_3 * \alpha \tag{13}$$

DFBCP$_{PC}$ computes the place of the controller according to $\tau$, which represents the propagation delay, $\rho$ represents the packet loss rate, and $\alpha$ represents the controller latency. Experiments were conducted to select the appropriate weights for $\varpi_1$, $\varpi_2$, and $\varpi_3$, as shown in Table 1.

**Table 1.** Selected values for the weights.

|   | $\varpi_1$ | $\varpi_2$ | $\varpi_3$ |
|---|---|---|---|
| 1 | 0.7 | 0.2 | 0.1 |
| 2 | 0.1 | 0.7 | 0.2 |
| 3 | 0.2 | 0.7 | 0.1 |

For each switch, a dynamic list of candidate controller places is constructed in descending order based on the DFBCP$_{PC}$ utility function. The dynamic list is updated through a feedback control theoretic technique mechanism. Switches start by connecting to the first controller in the list. All controllers' places are assumed to be reachable and accessible by all switches. In classical engineering environments, physical laws govern the relationships between the outputs and the inputs. This process is referred to as *first-principles techniques*. The main barrier for not using the first-principle modeling in the computing system domain is that some unrealistic assumptions are made. For that reason, using empirical approaches for developing transfer functions through the autoregressive moving average (ARMA) approach is more feasible for SDNs [64]. As shown in Figure 3, DFBCP$_{SDN}$ uses the classical control theoretic methodology. The classical control theoretic technique works in two stages: (1) the *system identification phase* and the (2) *controller design phase*.

In the system identification phase, transfer functions are constructed to model different system modules. DFBCP$_{SDN}$ uses the autoregressive moving average (ARMA) mathematical model for the system identification phase, and least square regression is used for ARMA parameter estimation. DFBCP$_{SDN}$ relies on having a tuning parameter that is easy to control. This tuning parameter has an influence on a controlled output parameter. The system desires to ultimately achieve a certain target value for the controlled output parameter. The controlled output parameter is a parameter that is needed to be controlled but cannot be adjusted directly, and hence the tuning parameter comes into place. The tuning parameter could be directly tuned and has an impact on the controlled output parameter.

In the controller design phase, a particular control law is used. Root locus is used to select the appropriate roots, hence finding the stable regions where the control law values

are selected. In the following sections, more discussion about the system identification phase and the control law phase used for the controller module is explained in more detail.

### 5.1. DFBCP$_{SDN}$ System Identification through the ARMA Model

DFBCP$_{SDN}$ performs the system identification through *Autoregressive Moving Average (ARMA)* model. System identification phase deals with linear regression to model SDN elements. The time domain of the ARMA model is expressed as shown in Equation (14). Equation (14) expresses the output of the element. The output in the time domain is referred to as $y(t)$, while the input in the time domain is referred to as $x(t)$.

We assumed $n$ series of old weighted values of the output and $m$ weighted values of input values, as shown in the ARMA model shown in Equation (14). The values of $i$ and $j$ are the index values of the parameters. Transfer functions are mathematical models that express the relationship between the outputs and the inputs. Equation (15) has the Z-transform equivalence for Equation (14). These transfer functions were used in the Z transform. The Z transform is considered as the frequency model of the time domain that is easier to deal with mathematically. The main goal is to design the appropriate gain that results in a stable system. The proposed model adds feedback that adds more control over the desired value of the controlled output parameter using the tuning parameter.

$$y(t) = \sum_{i=1}^{n} a_i \times y(t-i) + \sum_{i=0}^{m} b_j \times x(t-j) \tag{14}$$

In order to ease the mathematical modeling and derivation of the ARMA model, a frequency Z-domain version of ARMA is derived as shown in Equation (15).

$$H(z) = \frac{Y(z)}{X(z)} = \frac{\sum_{j=0}^{m} b_j \times z^{n-j}}{z^n - \left(\sum_{i=1}^{n} a_i \times z^{n-i}\right)} \tag{15}$$

The ARMA model in the time domain is applied to two modules: the place and the monitor modules as shown in Equation (16).

$$plc_{s_i,d_j}(t) = a_1 plc_{s_i d_j}(t-1) + b_0 pc_{s_i d_j}(t) \tag{16}$$

The variable *plc* represents the decision on the placement of the controller in the time domain. *plc* is a function in the tuning parameter *pc*. The ARMA model in the frequency domain for the place and the monitor modules is expressed in Equation (17).

$$PLACE_{s_i d_j}(z) = \frac{PLC_{s_i d_j}(z)}{PC_{s_i d_j}(z)} = \frac{b_0 z}{z - a_i} \tag{17}$$

### 5.2. DFBCP$_{SDN}$ Control Law and Gain Selection

DFBCP$_{SDN}$ uses the controller placement *pc* variable as the tuning parameter for placing the controller input of the control law, as expressed in Equation (18). The DFBCP$_{SDN}$ engine module uses the proportional integral (PI) controller. The distinguishing feature of the PI controller is the ability to use the two control terms of proportional and integral influence on the controller output to apply accurate and optimal control, as expressed in Equation (18).

$$pc_{s_i,d_j}(t) = pc_{s_i d_j}(t-1) + K_{s_i d_j} e_{s_i d_j}(t-1) \tag{18}$$

The error value is calculated as the difference between the reference value required by the master controller and the received values from the SDN's switches of the suggested parameters towards selecting the next master controller, as shown in Equation (19).

$$e_{s_i,d_j}(t) = ref_{s_i d_j}(t) - sc_{s_i d_j}(t) \tag{19}$$

In order to be able to compute a stable gain *K*, root locus is used for gain selection. The root locus plot is shown in Figure 4. The gain *K* is chosen to be 0.5 to obtain the closed-loop root of 0.59.

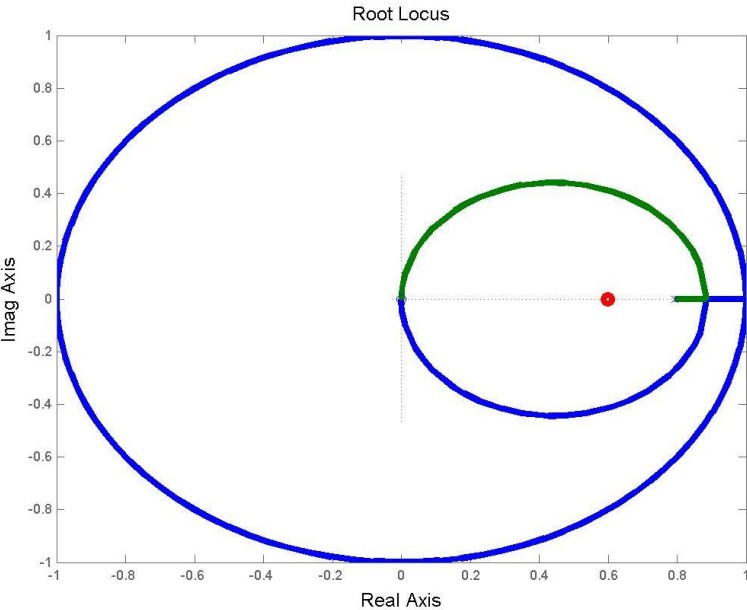

**Figure 4.** Root locus for the DFBCP model.

DFBCP$_\mathrm{SDN}$ uses least squares regression to estimate the values of the parameters of the ARMA model $a_1$, and $b_0$. $a_1$ is estimated to be 0.794 and $b_0$ is estimated to be 0.648. The goodness of the model is measured using $R^2$. $R^2$ is measured to be 89.7% as an indication of the linearity of the model. Equation (20) has the Z-transform equivalence for Equation (17) that describes the relationship between the selected backup to master controller and the tuning parameter.

$$PLACE_{s_i d_j}(z) = \frac{PLC_{s_i d_j}(z)}{PC_{s_i d_j}(z)} = \frac{b_0 z}{z - a_1} \tag{20}$$

The engine module transfer function for the control law that relates the master control to the appropriate tuning parameter is given in Equation (21) to model the SDN controller master module derived.

$$ENGINE_{s_i d_j}(z) = \frac{PC_{s_i d_j}(z)}{E_{s_i d_j}(z)} = \frac{K_{s_i d_j}}{z - 1} \tag{21}$$

The overall transfer function that relates the selected controller to the reference value could be given the transfer function stated in Equation (22).

$$T(z) = \frac{PLC(z)}{R(z)} = \frac{K \times z \times (z \times b_0)}{(z - 1) \times (z - a_1) + K \times z \times (z \times b_0)} \tag{22}$$

DFBCP$_\mathrm{SDN}$ uses least squares regression to estimate the values of the parameters of the ARMA model $a_1$, and $b_0$. $a_1$ is estimated to be 0.4364 and $b_0$ is estimated to be 0.2898. It is found that the $a_1$ that measures the goodness of the model is no lower than 87.5% for the controllers. The DFBCP$_\mathrm{SDN}$ algorithm is provided in Algorithm 1. The algorithm shown in Algorithm 1 describes the initialization process and the procedure to modify the transfer functions to adapt to the underlying dynamic system.

---

**Algorithm 1** DFBCP$_{SDN}$ Algorithm.

---

　　**Input:** $\omega$, reference target, $\eta$, Termination criteria
　　**Output:** FBPlacement, FBUtilityfunction

---

　1: **procedure** DFBCP$_{SDN}$(*FBPlacement*, *FBUtilityfunction*)
　2:　　**CLUSTER** = Initialize cluster of controllers for different classes
　3:　　$\eta$ = Number of switches in a cluster
　4:　　Randomly initialize each of the controllers as $\chi_1, \chi_2, \chi_3, \ldots \chi_N$ whole population
　5:　　Set *reference$_{target}$* to 90%
　6:　　Calculate the utility function $\omega$ for each controller
　7:　　**while** Termination criteria is not met **do**
　8:　　　　Sort controllers' locations in ascending order based on utility function
　9:　　　　$\chi_{best}$ = Get maximum utility function at the controller
　10:　　　　$\chi_{worst}$ = Get minimum utility function at the controller
　11:　　　　$\chi_{peer}$ = Get the utility function to the tentative controller placement
　12:　　**end while**
　13:　　**for** $i \leftarrow 1$ to $\eta$ do **do**
　14:　　　　$\chi_{peer}$ = randomly choose a controller position not in $\chi_i$
　15:　　　　Compute current utility function:
　16:　　　　$plc_{s_i, d_j}(t) = a_1 plc_{s_i d_j}(t-1) + b_0 pc_{s_i d_j}(t)$
　17:　　　　$pc_{s_i, d_j}(t) = pc_{s_i d_j}(t-1) + K_{s_i d_j} e_{s_i d_j}(t-1)$
　18:　　　　$e_{s_i, d_j}(t) = ref_{s_i d_j}(t) - sc_{s_i d_j}(t)$
　19:　　**end for**
　20:　　Compare $\chi_{peer}$ to the $\chi_{worst}$ and $\chi_{best}$
　21:　　**if** utility function$\leq$ *reference$_{target}$* **then**
　22:　　　　*Controller placement $\leftarrow$ succeeding available placement location from the list*
　23:　　**else**
　24:　　　　**if** *utilityfunction*>*reference$_{target}$* **then**
　25:　　　　　　*Controller placement $\leftarrow$ precedent available placement location from the list*
　26:　　　　**end if**
　27:　　**end if**
　28:　　**while** $COP - TARGET > THRESHOLD$ **do**
　29:　　　　FBPlacement = Get node having maximum utility function
　30:　　　　FBUtilityfunction = utility function corresponding to FBPlacement
　31:　　**end while**
　32:　　**return** FBPlacement, FBUtilityfunction
　33: **end procedure**

---

## 6. Results and Analysis

We used MATLAB as the platform for the simulation in addition to Python for clustering purposes. The system consists of Windows 10 (64-bit) with Intel Core i7-4770 CPU at 3.40 GHz and 16 GB RAM. For all experiments, we used a population size of 1000. In this section, we show the comparative results of both the proposed DFBCP$_{SDN}$ and the VBO reference models. Experimental results show that DFBCP$_{SDN}$ outperforms the VBO reference model in two scenarios, the Internet2 OS3E and EU-GÉANT [65]. The proposed DFBCP$_{SDN}$ model gives a higher convergence rate as compared with the VBO reference model. We show the cumulative results for both the ARMA-based solutions and VBO solutions for the CPP.

Figure 5 has the Internet2 OS3E topology, while Figure 6 has the EU-GÉANT topology. Both these topologies were used during the experimentation phase. We used a set of well-known network topologies for our experiments and used the DFBCP$_{SDN}$ and VBO algorithms for finding controller placements in a given topology. The topologies used are Internet2 OS3E topology and EU-GÉANT topology. Figure 7 shows the convergence graphs for the controller placement problem for the uncapacitated scenario for the Internet2 OS3E topology. Figure 7 has the relationship of the function of evaluation among the average rate of latency. DFBCP$_{SDN}$ outperformed the VBO by 11%. Figure 8 has the relationship

of the function of evaluation among the average rate of latency when using EU-GÉANT. DFBCP$_{SDN}$ outperformed the VBO by 9%. The average rate of latency drops at 30 switches is due to the saturation of switches around 30, so the total average latency saturates at that number, and then the more the number of switches, the trend moves on in an increasing fashion.

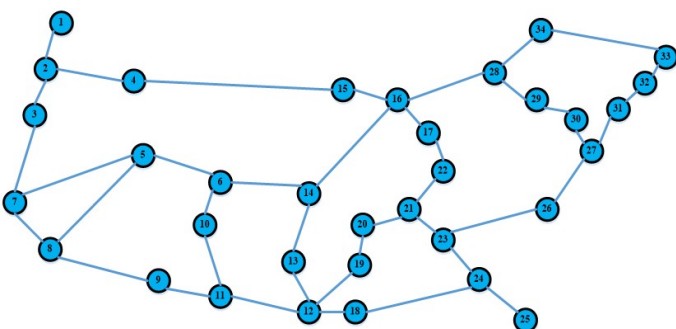

**Figure 5.** Internet2 OS3E topology.

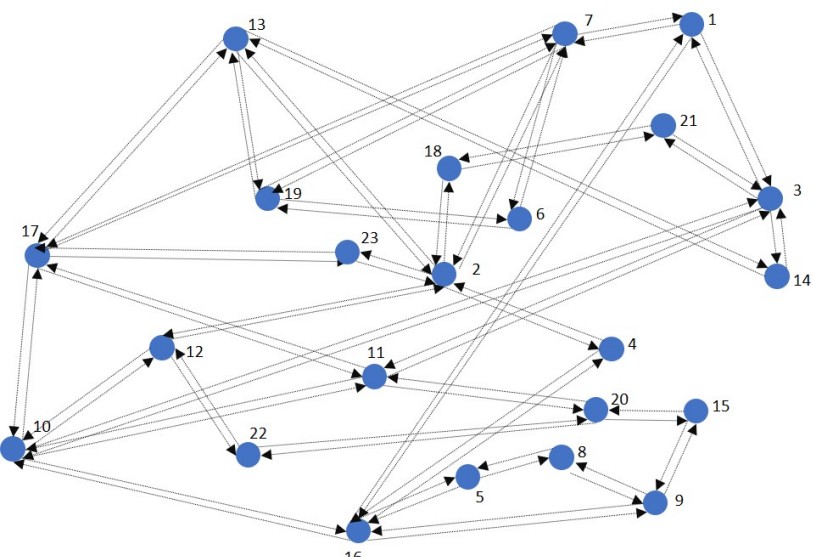

**Figure 6.** EU-GÉANT topology.

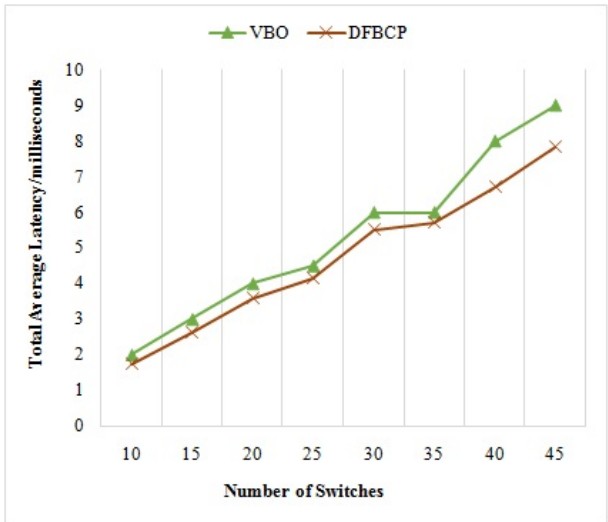

**Figure 7.** Convergence plots for uncapacitated controller placement problem for Internet2 OS3E topology.

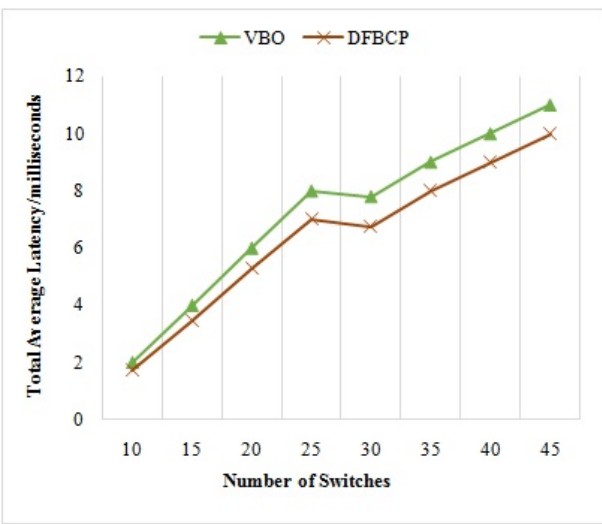

**Figure 8.** Convergence plots for uncapacitated controller placement problem for EU-GÉANT.

Figure 9 has the convergence plots for the controller placement problem for the capacitated scenario when used in an Internet2 OS3E topology. $DFBCP_{SDN}$ outperformed the VBO by 10%. Figure 10 has the convergence plots for the capacitated controller placement problem for EU-GÉANT topology. $DFBCP_{SDN}$ outperformed the VBO by 8%.

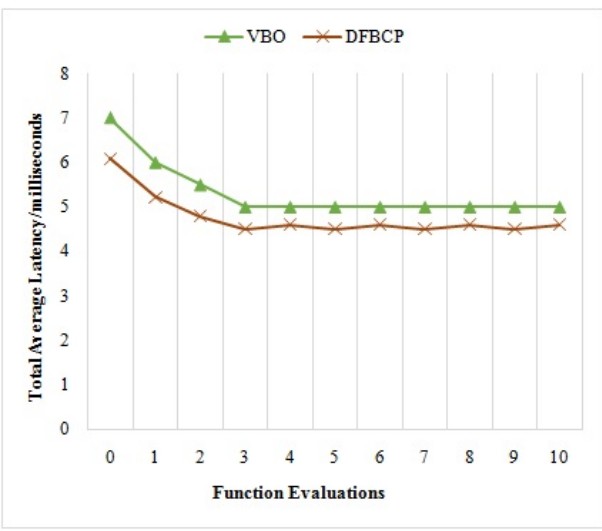

**Figure 9.** Convergence plots for capacitated controller placement problem for Internet2 OS3E topology.

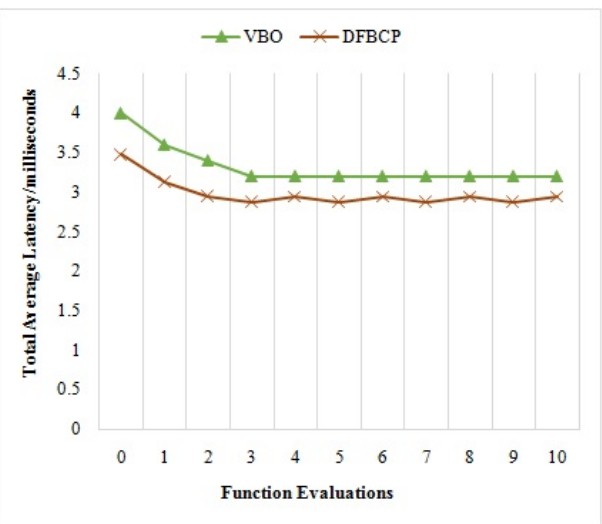

**Figure 10.** Convergence plots for capacitated controller placement problem for EU-GÉANT topology.

### 7. Conclusions and Future Work

Controller placement is an important problem in the large-scale SDN. An efficient controller placement solution attempts to minimize the total average latency of SDN network components to maximize overall SDN performance. This work discusses and analyzes the clustering-based solutions for controller placement. We propose a novel dynamic feedback control mechanism for controller placement; the proposed model is referred to as $DFBCP_{SDN}$. In this work, we conducted a set of experiments to compare the results of the $DFBCP_{SDN}$ with a reference model, which is the Varna-based optimization (VBO). The reason we selected the VBO is that the VBO is one of the latest algorithms that uses optimization techniques to reduce the overall latency. In previous work in the literature, the VBO has outperformed other models, and hence was a good candidate for our comparison purposes. In this work, we used different scenarios and topologies to conduct the comparison. The comparison was conducted in various scenarios using two topologies: the Internet2 OS3E topology and the EU-GÉANT topology. $DFBCP_{SDN}$ used feedback control theoretic techniques based on ARMA models. Experiments show that for the uncapacitated CPP, $DFBCP_{SDN}$ significantly outperforms the VBO for the Internet2 OS3E and EU-GÉANT topologies by 11% and 9%, respectively. Experiments also showed that for capacitated CPP, the $DFBCP_{SDN}$ algorithm outperforms the VBO reference model by 10% and 8%, respectively, in terms of total average latency.

A future work to this research is to add more artificial intelligent approaches to determine the controller placement criteria. Another direction is to test the model with more sophisticated topologies and more numbers of switches and controllers. A future work to add to this work is to measure more performance metrics. Load balancing among controllers is also considered a future aspect to this work.

**Author Contributions:** Formal Analysis and writing original draft, W.H.F.A.; Visualization, H.K.; Data curation, S.A.; Investigation, N.M.; Resources, K.S. All authors have read and agreed to the published version of the manuscript.

**Funding:** This research received no external funding.

**Conflicts of Interest:** The authors declare no conflict of interest.

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
