# Peer review of "Dynamic Feedback versus Varna-Based Techniques for SDN Controller Placement Problems"

_electronics, doi:10.3390/electronics11142273_

Round 1
Reviewer 1 Report
The authors have proposed a dynamic feedback algorithm for controller placement for SDNs (DFBCP) and compared it to the reference model, VBO, which is claimed to be a relatively new approach that is addressing the CPP. However, the recently-proposed literature as below are more relative to the proposed DFBCP model. The authors are recommended to include them in the Related Work and made a comparison. Besides, Figure 8 shows an upward trend for the average rate of latency when the number of switches increases but the average rate of latency drops at 30. No discussions are found about the phenomenon.
1. S. Mohanty, A. S. Shekhawat, B. Sahoo, H. K. Apat and P. Khare, "Minimizing Latency for Controller Placement Problem in SDN," 2021 19th OITS International Conference on Information Technology (OCIT), 2021, pp. 393-398.
2. V. Huang, G. Chen, Q. Fu and E. Wen, "Optimizing Controller Placement for Software-Defined Networks," 2019 IFIP/IEEE Symposium on Integrated Network and Service Management (IM), 2019, pp. 224-232.
3. P. Tao, C. Ying, Z. Sun, S. Tan, P. Wang and Z. Sun, "The Controller Placement of Software-Defined Networks Based on Minimum Delay and Load Balancing," 2018 IEEE 16th Intl Conf on Dependable, Autonomic and Secure Computing, 16th Intl Conf on Pervasive Intelligence and Computing, 4th Intl Conf on Big Data Intelligence and Computing and Cyber Science and Technology Congress(DASC/PiCom/DataCom/CyberSciTech), 2018, pp. 310-313.
Reviewer 2 Report
In this paper, the authors proposed a novel scheme for controller placement in SDN, which is an important problem regarding the software-defined networks. I have the following recommendations regarding improvements to the paper.
1. State clearly the novelty of your proposed scheme.
2. Add the problem statment a
3. Insert the Motivation of the proposed methodology
4. Put some recent literature published in last 2 years on controller placement issues such as in SD-IoT.
5. Add some future scope and directions of the research.
Round 2
Reviewer 1 Report
In lines 379-381 of the revised manuscript, authors have discussed why the curves in Figure 8 show an upward trend for the average rate of latency with the number of switches but the average rate of latency drops at 30. However, the discussion is concerned with Figure 8, lines 379-381 should not be a new paragraph.
